# Dual Mechanisms of Salinity Tolerance in Wheat Germplasm Lines W4909 and W4910

**DOI:** 10.3390/ijms252312892

**Published:** 2024-11-30

**Authors:** Richard R.-C. Wang, Steven S. Xu, Thomas A. Monaco, Matthew D. Robbins

**Affiliations:** 1Forage & Range Research Laboratory, USDA-ARS, Logan, UT 84322-6300, USA; thomas.monaco@usda.gov (T.A.M.); matthew.robbins@usda.gov (M.D.R.); 2Crop Improvement and Genetics Research, USDA-ARS, Albany, CA 94710, USA; steven.xu@usda.gov

**Keywords:** hexaploid wheat, *Triticum aestivum*, salt tolerance, sodium exclusion, tissue tolerance, specific locus-amplified fragment (SLAF), sequence-tagged sites (STS), molecular marker

## Abstract

Soil salinity adversely affects plant growth and development, reducing the yield of most crops, including wheat. The highly salt-tolerant wheat germplasm lines W4909 and W4910 were derived from a cross between two moderately salt-tolerant lines, the Chinese Spring (CS)/*Thinopyrum junceum* disomic addition line AJDAj5 (AJ) and the Ph-inhibitor line (Ph-I) derived from CS/*Aegilops speltoides*. Molecular markers for gene introgressions in W4909 and W4910 were not reported. Four sequence-tagged site (STS) molecular markers of Ph-I were developed and tested in the above-mentioned lines and the F_2_ progenies of the two crosses, Anza (AZ) × 4740 (sib of W4910) and Yecora Rojo (YR) × 4728 (sib of W4909). Additionally, homogeneity was assessed in several derivatives of W4909, 4728, W4910, and 4740 using the four markers. The four STS markers are not associated with salt tolerance, but they provide an indication of the transfer of chromatin in 3**B** chromosome of *Ae. speltoides* via Ph-I. Moreover, salt tolerance and leaf sodium concentration were determined in CS, AJ, Ph-I, 7151 (progeny of W4909), 7157 (progeny of W4910), AZ, and YR under salt treatment and control. Surprisingly, AJ had the lowest leaf sodium concentration under the control and salt treatment, indicating greater sodium exclusion than that in CS, AZ, and YR. This low level of leaf sodium concentration was heritable from 4740 to its hybrid progenies. On the other hand, the higher leaf sodium concentration, indicative of the tissue tolerance to salinity in Ph-I, had been inherited by both W4909 and W4910 and then transmitted to their hybrid progenies. One offspring line each in both W4909 and W4910 (7762 and 7159, respectively) were homozygous for the three molecular markers and lacked the marker psr1205 of *Su1-Ph1* gene, making them better materials than the original lines for future research on, for example, whole-genome sequencing and gene mining. The implications of these findings for the utilization of W4909 and W4910 in breeding salt-tolerant wheat cultivars are discussed.

## 1. Introduction

Due to a global change in climate, increasing soil salinization that affects soil health and constrains agricultural production has become a major land-degradation problem [1,2]. Worldwide, over 1 billion hectares of land suffer from salinization [3]. Soil salinity greatly affects bread wheat (*Triticum aestivum* L., 2*n* = 6*x* = 42, AABBDD) yield and quality; thus, enhancing the salt tolerance of wheat is a vital task to sustain wheat production for human consumption [4]. However, bread wheat has limited genetic variability in salt tolerance that can be broadened by introducing genes from species in the genus *Thinopyrum* of the Triticeae tribe, which are tolerant to salinity and can easily be hybridized with wheat.

Useful genes from wild Triticeae species can be transferred into tetraploid wheat (*Triticum durum* Desf., 2*n* = 4*x* = 28, AABB) and hexaploid wheat by inducing gene introgression through homoeologous chromosome pairing and suppressing or inhibiting the effect of the homologous pairing (Ph) gene *pairing homoeologous 1* (*Ph1*) on the long arm of 5**B** chromosome [5]. Other methods, including irradiation and tissue culture, entail several disadvantages, such as a genetic imbalance in translocation lines involving recombination between non-homoeologous chromosomes [6].

Data on chromosome pairing between wheat and *Ae. speltoides* Tausch. (2*n* = 2*x* = 14, **SS**) revealed natural variation in homoeologous pairing, resulting in low-, intermediate- and high-pairing hybrids [7,8,9,10]. The high-pairing hybrids between wheat containing *Ph1* and *Ae. speltoides* accessions indicated the presence of inhibitors or suppressors of *Ph1* in those *Ae. speltoides* accessions. Consequently, the genes inhibiting and suppressing the *Ph1* gene were transferred to hexaploid wheat [11,12,13]. The high-pairing Ph-inhibitor line (Ph-I) carrying the genes *Ph^I^* was tentatively identified to be a translocation line involving 4**D**/4**S**, based on chromosome C-banding, and another unidentified chromosome pair [9]. Alternatively, the suppressor genes *Su1-Ph1* and *Su2-Ph1* were mapped to the 3**S** and 7**S** chromosome of *Ae. speltoides*, respectively [12]. The genome symbol for *Ae. speltoides* was recently changed from **S** to **B** [14,15], as suggested nearly 30 years ago [10], such that the chromosomes of *Ae. speltoides* will be written hereafter as 1**B** to 7**B** of *Ae. speltoides*.

The Ph-I line was crossed with AJDAj5, the Chinese Spring (CS)/*Thinopyrum junceum* disomic addition line, to produce translocation lines W4909 and its sib line 4728, and W4910 and its sib line 4740, which were tolerant of salinity up to EC = 42 dS/m [16]. These lines inherited salt tolerance from both the parental lines AJDAj5 and Ph-I, characterized by having an extremely high leaf sodium concentration, in contrast to the sodium exclusion mechanism that results in low leaf sodium concentrations. Thus, W4909 and W4910 were released as wheat germplasm lines [17]. The tissue tolerance to salinity of W4909 was substantiated and used to develop a high-yielding germplasm line MW#293 tolerant of both salinity and sodicity [18]. It was advocated that tissue tolerance of salinity would be the basis for breeding salt-tolerant wheat cultivars in the future [16]. Other than Ph-I, W4909, W4910, and MW#293, only one Portuguese landrace, Mocho de Espiga Branca, accumulates up to sixfold greater leaf and sheath sodium concentrations than two Australian cultivars, Gladius and Scout [19].

The State of California in the U.S. has a vast acreage of saline soil [3]. Wheat cultivars Yecora Rojo (YR) and Anza (AZ) are well adapted to California wheat production. Therefore, before the release of W4909 and W4910, their sib lines 4728 and 4740 were used to cross with YR and AZ, respectively.

Because the tissue tolerance of salinity was contributed from the Ph-I, it is logical to assume that gene (or genes) conferring this tolerance mechanism is (or are) located on the chromosomes transferred from *Ae. speltoides* accession TA1786 into Ph-I [11]. *Xpsr1205*, a molecular marker 0.4 cM distal to *Su1-Ph1* [12] on 3**B** of *Ae. speltoides*, was used to test whether *Xpsr1205* is present in Ph-I, W4909 or W4910. Additionally, three specific locus-amplified fragment sequencing (SLAF) sequences were converted into sequence-tagged sites (STS) markers and tested on parental lines and F_2_ hybrid derivatives of 4728 (sib of W4909) and 4740 (sib of W4910). STS markers would be useful in the marker-assisted selection (MAS) of breeding materials generated via crosses involving W4909 and/or W4910. This study aimed to achieve the following: (1) the development of STS molecular markers that are present in AJDAj5 or Ph-I and W4909 or W4910 but absent from Chinese Spring, and testing these STS markers to identify newer lines that lack the marker psr1205 for the Su*1-Ph1* gene on the 3B chromosome of *Ae. speltoides*; and (2) the determination of the leaf sodium concentration in parental lines and F_2_ populations of the YR × 4728 and AZ × 4740, to elucidate the mechanisms of salinity tolerance. The results of this study reveal genetic lines without the potential for chromosome instability in hybrid derivatives and show that the two target lines have differing mechanisms for salt response.

## 2. Results

The plant materials used in this study are shown in Figure 1. Because AJDAj5 and the Ph-inhibitor line Ph-I were both developed in the CS background [11,20], W4909 and W4910 also shared the CS background. These five lines were used to generate specific locus-amplified fragment (SLAF) sequences.

### 2.1. Development of SLAF-Derived STS Markers for W4909 and W4910

SLAF sequences were generated from DNA samples of CS, AJ, Ph-I, W4909, and W4910 and provided by Dr. Xingfeng Li of Shandong Agricultural University, Tai’an, China. An analysis of SLAF sequences yielded some potential markers of W4909 and/or W4910 (Table 1)

Despite testing 218 primer pairs designed from 156 SLAF sequences on CS, AJ, Ph-I, 7151 (progeny of W4909), 7157 (progeny of W4710), YR, and AZ, only four STS markers were successfully developed for W4909 and W4910 (Table 2). Additional molecular markers could be developed from other sequences identified as polymorphic between CS and W4909/W4910 (Appendix A). The results of PCR amplification for Marker4607154 in parental lines and hybrid progenies were the same as those of Marker264410. Therefore, only the results of Marker264410 were presented. The random fragment length polymorphism (RFLP)-derived STS marker psr1205 was previously reported as a marker for the *Ph1* suppressor gene, *Su1-Ph1* [12]. Now, it is shown that this marker also exists in the *Ph1* inhibitor line Ph-I, 7151 (progeny of W4909) and F_2_ segregants of the YR × 4728 cross (Figure 2). The SLAF-derived STS Marker264410 was present in Ph-I, W4910, YR, and many F_2_ segregants of YR × 4728 (Figure 3) but absent from F_2_ segregants of AZ × 4740 (Table 3). The genotype of markers psr1205 and Marker264410 in the 62 out of 66 F_2_ plants from the YR × 4728 cross suggests that the two molecular markers had a genetic distance of (4 + 21)/(4 + 21 + 17 + 20) = 40.33 centimorgan (cM) (Table 3) and a physical distance of 17.22 Mb (Table 2). The SLAF-derived STS Marker6805321 showed an identical pattern to Marker264410 in Ph-I and line 7157 (progeny of W4910) but was absent in F_2_ populations (lines 5722 and 5758) of both AZ × 4740 and YR × 4728 crosses (Table 3). Marker6805321 was present in F_1_ of YR × 4728-y (line 5445 in Table 4) and progenies of W4909 and W4910 (lines 7284, and 5978, 7159, 7285 in Table 5; 7159 in Figure 4).

As 4728 and 4740 were sibs of W4909 and W4910, respectively, the results for Marker264410 and Marker6805321 were puzzling; i.e., these two markers were present in 7157 (progeny of W4910) but absent from all F_2_ individuals of the cross AZ × 4740 (sib of W4910) (Table 3). Because seeds of lines 4728 and 4740 lost germinability due to long seed storage, we could only check the molecular marker profiles in F_1_ hybrids of 4728 crossed with Anza and Yecora Rojo (Table 4), as well as other progeny lines of W4909 and W4910 (Table 5), to infer their molecular genotypes.

**Table 2 ijms-25-12892-t002:** Primer sequences of working STS markers derived from RFLP marker psr1205 and SLAF sequences 264410, 4607154, and 6805321.

Marker	Chromosome Location	Primer Name	Primer Sequence 5′ --> 3′	T_A_ °C	Amplicon bp
Psr1205	chr3B:541288907..541289382	Forward	CGGCAATGATGAGTGTGTCAT	59 to 56	362
		Reverse	CAACTCCCAGTTTGCTGACA		
Marker264410	chr3B:558511365..558511884	Forward	ACACTACTCATACGGAACCATCG	55	462
		Reverse	TCTTGGCTGACTTGGCATTCA		
Marker4607154	chr3B:558833142..558833625	Forward	ACAAGCAACTAACAGAGCCA	55	486
		Reverse	CTGTCGATGCAGGGTTCTACT		
Marker6805321	chr2E:49152430..49152911	Forward	AATGTGAACAATCAACGAGATGT	52	377
		Reverse	GTGCACAACACACAGTGGTC		

Apparently, plants in line 4728 used in the crosses with Anza and Yecora Rojo could be classified as two genotypes. Type 4728-x was heterozygous for psr1205 and homozygous-negative for Marker264410 and Marker6805321, while the type 4728-y plants were probably negative for psr1205 and homozygous positive for Marker264410 and Marker6805321 (Table 4; Appendix A). The plant 4728-x was the parent in the crosses AZ × 4728 (lines 5440 and 5441) and YR × 4728 (lines 5446 and 5447) (Table 4). The F_1_ hybrid of Yecora Rojo × 4728-x, that gave rise to the F_2_ population 5758 (Appendix A) would be heterozygous for psr1205, heterozygous for M264410 that came from YR, and homozygous-negative for M6805321 (Table 3). Only the segregation ratios of 38:24 for Marker264410 (Table 3) deviated from the ratio of 3:1 in this F_2_ population.

**Table 3 ijms-25-12892-t003:** Molecular markers in grandparents, parents, and F2 populations of Anza × 4740 and Yecora Rojo × 4728. Green + indicates Yecora Rojo as the source of Marker264410 in the F_2_ population.

ID ^1^	Description	No. Plants	psr1205	Marker264410	Marker6805321
6687	Chinese Spring	8	-	-	-
6598	AJDAj5	7	-	-	-
6621	Ph inhibitor	8	+	+	+
7151	Progeny of W4909	8	+	-	-
7157	Progeny of W4910	9	-	+	+
5976	Yecora Rojo	9	-	+	-
5975	Anza	9	-	-	-
5722	(Anza × 4740) F_2_	91	-	-	-
	Total	91			
5758	(Yecora Rojo × 4728-x) F_2_	4	-	-	-
		21	+	+	-
		17	-	+	-
		20	+	-	-
	Total	62			

**^1^** ID numbers are seed packet numbers, each of them contains the seed harvested from one spike of a plant.

**Table 4 ijms-25-12892-t004:** Molecular markers in parental lines and F1 hybrids of 4728 (a sib of W4909). Green + indicates Yecora Rojo as the source of Marker264410 in the F_2_ population. Yellow + indicates Ph-I and 7151 or 7157 as the source of markers.

ID ^1^	Kind	psr1205	Marker264410	Marker6805321
6687	Chinese Spring	-	-	-
6598	AJDAj5	-	-	-
6621	Ph inhibitor	+	+	+
7151	Progeny of W4909	+	-	-
7157	Progeny of W4910	-	+	+
5975	Anza	-	-	-
5976	Yecora Rojo	-	+	-
5440-1	(Anza × 4728-x) F_1_	-	-	-
5440-2	(Anza × 4728-x) F_1_	+	-	-
5440-3	(Anza × 4728-x) F_1_	-	-	-
5440-4	(Anza × 4728-x) F_1_	-	-	-
5440-5	(Anza × 4728-x) F_1_	+	-	-
5440-6	(Anza × 4728-x) F_1_	+	-	-
5440-7	(Anza × 4728-x) F_1_	+	-	-
5440-8	(Anza × 4728-x) F_1_	+	-	-
5441-1	(Anza × 4728-x) F_1_	+	-	-
5441-2	(Anza × 4728-x) F_1_	-	-	-
5441-3	(Anza × 4728-x) F_1_	+	-	-
5441-4	(Anza × 4728-x) F_1_	-	-	-
5441-5	(Anza × 4728-x) F_1_	-	-	-
5441-6	(Anza × 4728-x) F_1_	+	-	-
5445-1	(Yecora Rojo × 4728-y) F_1_	-	+	+
5445-2	(Yecora Rojo × 4728-y) F_1_	-	+	+
5445-3	(Yecora Rojo × 4728-y) F_1_	-	+	+
5445-4	(Yecora Rojo × 4728-y) F_1_	-	+	+
5445-5	(Yecora Rojo × 4728-y) F_1_	-	+	+
5446-1	(Yecora Rojo × 4728-x) F_1_	+	+	-
5446-2	(Yecora Rojo × 4728-x) F_1_	-	+	-
5446-3	(Yecora Rojo × 4728-x) F_1_	+	+	-
5446-4	(Yecora Rojo × 4728-x) F_1_	-	+	-
5446-5	(Yecora Rojo × 4728-x) F_1_	-	+	-
5446-6	(Yecora Rojo × 4728-x) F_1_	+	+	-
5446-7	(Yecora Rojo × 4728-x) F_1_	-	+	-
5446-8	(Yecora Rojo × 4728-x) F_1_	-	+	-
5447-1	(Yecora Rojo × 4728-x) F_1_	-	+	-
5447-2	(Yecora Rojo × 4728-x) F_1_	-	+	-
5447-3	(Yecora Rojo × 4728-x) F_1_	+	+	-
5447-4	(Yecora Rojo × 4728-x) F_1_	-	+	-
5447-5	(Yecora Rojo × 4728-x) F_1_	-	+	-
5447-6	(Yecora Rojo × 4728-x) F_1_	-	+	-
5447-7	(Yecora Rojo × 4728-x) F_1_	-	+	-

^1^ ID numbers are seed packet numbers, each of them contains the seed harvested from one spike of a plant.

### 2.2. Homozygosity in Progenies of W4909 and W4910

The molecular-marker profiles of six progeny lines of W4909 and W4910, two to three generations of self-pollination from the uniformly salt-tolerant W4909 and W4910, were unknown. Therefore, they were assessed for homozygosity of the three STS markers in 16 to 20 plants per line. Lines 7762 and 7159 are homozygous for the three STS markers (Table 5). Line 7762 lacked markers M264410 and M6805321, while line 7159 included both. Both lines lacked the psr1205, which is the marker for the *Su1-Ph1* gene that could cause the chromosome instability of hybrids of W4909 and W4910, arising from homoeologous chromosome pairing. Therefore, lines 7762 and 7159 are more desirable than the original W4909 and W4910 for crossing with wheat cultivars to breed salt-tolerant wheat.

**Table 5 ijms-25-12892-t005:** Molecular markers in different progeny lines of W4909 and W4910. ID numbers are seed packet numbers; each contains seed harvested from a single plant. Red + indicate markers originate from Ph-I and 7151 or 7157; yellow blocks are homozygous markers in progeny lines of W4909 and W4910.

ID	Kind	psr1205	Marker264410	Marker6805321
CS	Chinese Spring	-	-	-
AJ	AJDAj5	-	-	-
Ph-I	Ph inhibitor	+	+	+
7151	Progeny of W4909	+	-	-
7157	Progeny of W4910	-	+	+
AZ	Anza	-	-	-
YR	Yecora Rojo	-	+	-
NC	Negative control	-	-	-
5977	Progeny of W4909	8 : 12 + : -	0 : 20 + : -	0 : 20 + : -
7284	Progeny of W4909	0 : 20 + : -	1 : 19 +: -	1 : 19 + : -
7762	Progeny of W4909	0 : 16 + : -	0 : 16 + : -	0 : 16 + : -
5978	Progeny of W4910	0 : 20 + : -	17 : 3 + : -	18 : 2 + : -
7159	Progeny of W4910	0 : 18 + : -	18 : 0 + : -	18 : 0 + : -
7285	Progeny of W4910	0 : 20 + : -	18 : 2 + : -	20 : 0 + : -

### 2.3. Survival, Leaf Sodium Concentration in Parents, and Progenies of W4909 and W4910

AJDAj5 had the lowest leaf sodium ion concentration in both control and salt-treated plants, although that in salt-treated plants was not statistically significantly different from CS, YR, and AZ (Table 6). Ph-I, W4909, and W4910 had significantly higher leaf sodium concentrations than the other four lines under both the control and salt-stressed conditions. The survival rate, leaf sodium concentrations, and marker profiles are presented in Appendix A. When percentage survival at a 52-day cutoff between tissue tolerance and susceptibility to salinity was used, AJDAj5, Ph-I, W4909, and W4910 were all more tolerant to salt stress than CS, AZ, and YR.

In addition to parental materials, the survival rate in terms of survival days since treatment (SDST) was measured on F_2_ plants from AZ × 4740 and YR × 4728 crosses (Appendix A), and the leaf sodium ion concentration was measured in randomly selected F_2_ plants (24 of 95 and 14 of 66, respectively) of those two crosses (Figure 5). In the AZ × 4740 cross, 20 salt-tolerant F_2_ plants out of 24 measured plants had low leaf sodium concentrations, a trait apparently inherited from AJDAj5. On the other hand, 6 out of 10 salt-tolerant F_2_ plants in the YR × 4728 cross had a high sodium concentration in leaves, a trait attributable to Ph-I. However, a high leaf sodium concentration alone could not confer or account for salt tolerance in F_2_ segregants in both crosses, such as plants 16, 31, and 46 in YR × 4728 and plants 120, 146, and 170 in AZ × 4740. There is, however, strong evidence (*p*-value of the Welch two-sample *t*-test = 0.003529) in the AZ × 4740 cross that salt-tolerant F_2_ individuals had lower sodium ion concentrations. This supports the conclusion that the mechanism from 4740 (sib of W4910) is sodium exclusion, but tissue tolerance was contributed from 4728 (sib of W4909) in the YR × 4728 cross.

### 2.4. Segregation of Salt Tolerance in F_2_ Populations of the Two Crosses, Anza × 4740 and Yecora Rojo × 4728

F_2_ individuals of the crosses AZ × 4740 and YR × 4728 segregated in a 3:1 and 7:9 ratio, respectively (Table 7). Thus, a single dominant gene conferred salt tolerance in the hybrid derivatives of AZ × 4740, and two supplementary recessive genes controlled the trait in YR × 4728. STS markers in these two F_2_ populations (Appendix A) did not segregate at these ratios, indicating that these molecular markers were not associated with genes for salt tolerance. Again, these markers are merely indications of the presence of chromatin from the 3**B** chromosome of *Ae. speltoides*.

## 3. Discussion

Both AJDAj5 [20] and Ph-I [9] were developed in the CS background with different alien chromosomes of *Th. junceum* and *Ae. speltoides*, respectively. The strong band of STS marker psr1205 was present in Ph-I and 7151 (progeny of W4909), as well as some F_2_ segregants in the YR × 4728 cross (Figure 2). Thus, the intense band possibly originated from the 3**B** chromosome of *Ae. speltoides*. This result leads to both implications and applications. Firstly, the presence of marker psr1205 in Ph-I line suggests that Ph-I contains the 3**B** chromosome of *Ae. speltoides*, as reported previously [21]. If the *Ph^I^* gene is indeed located on 4**B** involving in the 4**D/4S** (=4**D/4B**) translocation chromosome, as reported earlier [11], then the unidentified chromosome in the Ph-I line would be the 3**B** chromosome of *Ae. speltoides*. Secondly, if both *Su1-Ph1* and *Ph^I^* are located on the 3**B** chromosome of *Ae. speltoides,* shown in a previous study [21] and confirmed in the current study, it raises the possibility that *Ph^I^* and *Su1-Ph1* are either different alleles of the same gene or two different genes located on the same chromosome. These possibilities call for future experiments, such as testcrossing them with the same plant material that has alien chromosomes and *Ph1* to determine any differences between the progenies. Also, the genetic and physical distances between *Ph^I^* and *Su1-Ph1* should be assessed to determine whether they are different genes located on the same chromosome. The data in the previous study [21] indicated that *wmc674* and *wmc505* were on the short arm of *Ae. speltoides* 3**B** chromosome, whereas the three markers in the present study were near the distal end of 3**B** long arm. Despite the two studies confirming that the Ph-I line contained segments of both 3**B**S and 3**B**L of *Ae. speltoides*, the precise position of *Ph^I^* gene is still unknown.

An annotation of the 3**B** chromosome of *Ae. speltoides* spanning 13 Mb revealed many candidate genes for salt tolerance near markers psr1205 and Marker264410 reported in this study (Appendix A). Some of them had been implicated in the transcriptome study of Chinese Spring, AJDAj5, Ph-I, W4909, and W4910 [22]. These include genes encoding peroxidase, Ser-Thr protein kinase, Myb transcription factor, late embryogenesis abundant protein LEA_2, glutathione S-transferase, calmodulin-binding domain, calcium-dependent vacuole membrane protein, and MIP aquaporin, etc. Although the STS markers identified in this study are not associated with salt tolerance, DNA sequences on 3B chromosome of *Ae. speltoides* flanking the candidate salt-tolerance genes could be developed as molecular markers useful in MAS for breeding salt-tolerant wheat cultivars.

Na^+^ exclusion from leaves is associated with salt tolerance in cereal crops, including durum wheat [23,24], bread wheat [25,26,27], and wild relatives such as *Hordeum* species [28], tall wheatgrass [29], and *Triticum tauschii* [30]. The bread wheat cultivars ‘Berkut’ and ‘Krichauff’ had Na^+^ concentration (mg kg^−1^ DW) of 6308 ± 296 and 5942 ± 442, respectively, while the double haploid lines derived from the hybrid of these two parents had a corresponding value ranging from 2850 to 9733 [31]. Of the five QTL identified for Na^+^ exclusion, two were co-located with seedling biomass on chromosomes 2A and 6A. The 2A QTL appears to coincide with the previously reported Na^+^ exclusion locus in durum wheat that hosts one active *HKT1*;*4* (*Nax1*) and one inactive *HKT1*;*4* gene. Their measurements were comparable to those of CS, YR, and AZ in this study. Fourteen of the twenty salt-tolerant plants in the F_2_ population of AZ × 4740 had leaf sodium concentrations less than the lowest value of 2850 mg kg^−1^ DW observed in the doubled-haploid population [31]. The strong sodium-exclusion gene was inherited from AJDAj5 that can be traced to *Thinopyrum junceum* [16,20]. This gene has not been identified or mapped; thus, no new gene name is given here. Future research is needed to identify and isolate this gene.

Genes controlling sodium exclusion in wheat, *Nax1* and *Nax2*, played a significant role in wheat breeding for salt tolerance [32]. *Nax1*, which accounted for 38% of the phenotypic variation for a low Na^+^ concentration in leaf blades, was mapped to the long arm of chromosome 2A via a quantitative trait locus (QTL) analysis [33]. It was identified through fine mapping as an Na+ transporter of the HKT gene family HKT7 (HKT1;4) [34]. *Nax2* was previously located at chromosome 5**A** of *T. monococcum* and identified as HKT1;5 [35]. However, it was not present in 4A of *T. uratu* and *T. aestivum* but mapped to 4B and 4D [36].

The plasma membrane sodium/proton exchanger salt-overly-sensitive 1 (SOS1) is a critical Na^+^ efflux protein in plants. Gao et al. (2023) cloned three homologues of the *TaSOS1* gene in bread wheat, designated as *TaSOS1-A1*, *TaSOS1-B1*, and *TaSOS1-D1*, respectively, according to the location on group 3 chromosomes 3A, 3B, and 3D [37]. Other transporters that may allow the influx of sodium ions have been reported, such as AtPIP2;1 (aquaporin) [37]. In the wheat microarray study of Mott and Wang (2007) [22], the tonoplast aquaporin (Ta.21042; TC205156) was expressed at a higher level in the leaf of Ph-I, W4909, and W4910 than in CS and AJDAj5 under both the control and salt treatments. Another one, aquaporin (TaAffx.8804), appeared to be common to AJDAj5, W4909, and W4910. A potassium-channel protein (Ta.25613) was expressed intermediately in AJDAj5, W4909, and W4910 between CS and Ph-I. Another potassium-channel protein (TaAffx.56132) was lower in AJDAj5, W4909, and W4910 than CS and Ph-I.

Using the normalized salt stress-specific expression datasets developed by Mott and Wang (2007) [22], Mehta et al. (2021) studied the shoot and root tissue-specific expression of the identified genes during the tillering stage [38]. K^+^/Na^+^ selectivity in wheat under salt stress was enhanced via *Lophopyrum elongatum* chromosome arms 1ES, 7ES, and 7EL [39]. Recently, both 7E from *Th. elongatum* (2n = 14; EE) and 7E_1_L of *Th. ponticum* (2n = 70; EEEEEEEEEE) were shown to greatly mitigate the effects of salt stress on root and leaf growth [40]. 7E and 7E_1_L also enhanced the ability of plants to neutralize ROS and limit their harmful effects via the presence of efficient scavenging systems, involving enzymatic and non-enzymatic antioxidants, including superoxide dismutase (SOD), catalase (CAT), peroxidase (POD), and ascorbate peroxidase (APX) enzymes, as well as ascorbate [40]. Some of these enzymes had been implicated in the microarray study of W4909 and W4910 [22]. These reports support our observation that both W4909 and W4910 had SLAF sequences traceable to the 7E chromosome that carries many candidate genes for salt tolerance (Appendix A). Unfortunately, those SLAF sequences were not successfully converted to STS markers in this study. In the future, those SLAF sequences might be converted to other SNP molecular markers.

The genetic distance between molecular markers psr1205 and Marker264410 was fairly large, 40.33 cM. This fact could explain why the marker psr1205 was easily eliminated from the offspring of W4909 (Table 5). The *Su1-Ph1* gene linked to psr1205 can lead to chromosome instability, resulting in a low seed set. By selecting a high seed yield in the hybrid progenies of W4909, psr1205 and the *Su1-Ph1* gene could be eliminated.

## 4. Materials and Methods

### 4.1. Plant Materials

The plant materials used in this study (Figure 1) included bread wheat cultivars Chinese Spring (CS), Anza (AZ), and Yecora Rojo (YR), as well as germplasm lines AJDAj5 (AJ), Ph-inhibitor line (Ph-I), W4909, and W4910. Sibs and offspring of W4909 and W4910, which had uniform salinity tolerance for two self-pollination generations from lines 2407 and 2457, respectively [17], were also analyzed for the three STS markers to select lines lacking psr1205. F_1_ and F_2_ of YR and AZ crossed by 4728 (sib of W4909) and F_2_ of AZ × 4740 (sib of W4910) were analyzed to ascertain their molecular marker profiles. Seeds of the above wheat lines were stored in a refrigerator at 2 °C prior to various studies.

### 4.2. Greenhouse Study of Salt Tolerance

This study was conducted at a greenhouse located at the USDA-ARS, FRR, on the campus of Utah State University, Logan. The experiment was conducted using a completely randomized split-plot design with two parts to test the analyzed materials’ salt tolerance. The first part involved testing CS, AJ, Ph-I, W4909, W4910, YR, and AZ under control and salt treatment. The second entailed testing the F_2_ progenies of YR × 4728 and AZ × 4740 for salt tolerance under salt treatment. One hundred and forty plastic stadium cups (900 mL capacity; without drainage) were used for the first testing, which consisted of 10 cups each for the control and salt treatment of seven lines. Within the control and salt-treatment plot, the 70 cups were randomly arranged. In the second part, up to one hundred cups each were used to accommodate the F_2_ progenies of YR × 4728 and AZ × 4740. A single seed of the plant materials was directly placed into the silica sand-filled cup in this study. Seventy-seven and one hundred seeds were planted, but 66 and 95 seedlings were established for the YR × 4728 and AZ × 4740 F_2_ populations, respectively. All seedlings of the F_2_ populations were treated with salt solution the same way as the parental lines in the first part of the experiment.

Plants were grown for 80 days under ambient solar radiation, while the air temperature and relative humidity remained relatively stable at 27 °C and 35%, respectively. Each cup received 50 mL of water-soluble nutrient solution (20-20-20 NPK with micronutrients; Scotts Miracle-Gro Products Inc., Marysville, OH, USA) and was irrigated with deionized water daily to maintain field capacity, i.e., 11.5% soil-water content. Gravimetric soil-water content was determined by weighing individual containers on an electronic microbalance and adding water as needed to reach field capacity. Salinity treatments were imposed via watering with a saline solution of EC = 3 dS/m once a week for 8 weeks, starting when the plants’ fourth leaves had developed. Salt tolerance was determined using survival days since the treatment (SDST) of each plant [41]. Plants with SDST greater than 52 were classified as salt-tolerant. Salt-sensitive plants died between 42 and 52 days after the first salt treatment.

### 4.3. Molecular Characterization of Parental Lines and Hybrid Progenies

DNA was extracted from the third leaf of these plants using the QIAGEN DNeasy kit, following the manufacturer’s protocol (Germantown, MD, USA. Three DNA samples each from CS, AJ, Ph-I, W4909, and W4910 were used to generate SLAF sequences. Each SLAF sequence is composed of two 100-bp DNA sequences flanking unknown nucleotides of a varying length, which was represented as (N)_10_. The sequence of each SLAF marker was analyzed using the BLAST function against the whole-genome sequences of Chinese Spring v2.1 [42], *Thinopyrum elongatum* v1.0 [43], and *Aegilops speltoides* TS01 [15] using WheatOmics JBrowse (http://wheatomics.sdau.edu.cn) [44]. If marker sequences revealed an identity closer to the alien sequences than to the CS sequences, they were classified as either *Th. Junceum*-originated or *Ae. speltoides*-originated. These markers were aligned with their homoeologous wheat sequences to find variable sites that could be used for primer design at the website https://www.ncbi.nlm.nih.gov/tools/primer-blast/, 27 November 2024.

The primer sequences for *Xpsr1205* and SLAF-derived STS markers are listed in Table 2. The complex PCR condition for *Xpsr1205* was provided by Dr. Karin Deal, UC-Davis, as follow: 96° 5 m, 8× (94 °C 30 s 59 °C 30 s 68 °C 1 min), 8× (94 °C 30 s 57 °C 30 s 68 °C 1 min), 20× (94 °C 30 s 56 °C 30 s 68 °C 1 min), 68 °C 5 min, and 10 °C indefinite. The PCR reagent (25 µL) contained NEB (New England Biolabs, Ipswich, MA, USA) standard Taq polymerase 5 µL, primer pair 1 µL, and template DNA 1 µL, and 18 µL of double-distilled water (ddH_2_O) was used in the PCR of *Xpsr1205*. PCR for the Marker264410 and Marker6407154 was carried out using the reagent containing 5X GoTaq flexi 2 µL, GoTaq Flexi G2 Hot Start Polymerase 0.05 µL, primer pair 1 µL, template DNA 1 µL, dNTPs (25 mM) 1 µL, MgCl_2_ 1 µL, and ddH_2_O with the program set for initial denaturing at 95 °C for 2 min, 30 cycles of denaturing at 95 °C for 30 s, annealing at 55 °C for 30 s, and extension at 72 °C for 1 min, with the final extension at 72 °C for 5 min and the plate held at 4 °C for infinity. PCR products were separated in 1% agarose gels and photographed.

### 4.4. Molecular Markers in Offspring of W4909 and W4910

Sixteen to twenty plants of lines 5977, 7151, 7284, and 7762 from the lineage of W4909, as well as 5978, 7157, 7159, and 7285 from the lineage of W4910, were grown in pots. Plants of these lines were analyzed for the uniformity of the three STS markers, psr1205, Marker264410, and Marker6805321. DNA extraction and PCR conditions were the same as previously described.

### 4.5. Determination of Sodium Ion Concentration in Leaves

Three randomly selected plants of CS. AJ, Ph-I. W4909, W4910, YR, and AZ, as well as fourteen and twenty-four randomly selected plants from sixty-six and ninety-five established plants in F_2_ of YR × 4728 and AZ × 4740, respectively, were used for leaf sodium ion analysis. The penultimate leaf (the leaf just below the flag leaf) was collected from three each of the control and salt-treated plants of CS, AJ, Ph-I, W4909, W4910, YR, and AZ. The leaf sodium ion (Na^+^) concentration (mg Kg^−1^ DW) was determined in three dried leaf samples per line per treatment via inductively coupled plasma mass spectrometry (ICP-MS) at the Utah State University Analytical Laboratory. The mean sodium ion concentration of parental wheat lines was statistically tested using a one-way ANOVA at the *p* = 0.05 level.

## 5. Conclusions

In this study, we presented data showing that genes for sodium exclusion and tissue tolerance were traceable to AJDAj5 and Ph-I via lines 4740 and 4728, sib lines of W4910 and W4909, respectively. A strong band of the STS marker psr1205 located to the 3**B** chromosome of *Aegilops speltoides* was attributed to the Ph-I line. Three additional SLAF sequence-derived STS markers were also identified in Ph-I and some progenies of W4909 and W4910. The heterogeneity of molecular markers was observed in the early generations of W4909 and W4910, resulting in variations in molecular marker profiles among F_1_ hybrids. One homozygous line each (line 7762 and line 7159) in the progenies of W4909 and W4910, respectively, were identified as desirable plant materials for future research, such as whole-genome sequencing and mining for genes conferring salt tolerance, due to their lacking the marker psr1205 of *Su1-Ph1* gene.

## Figures and Tables

**Figure 1 ijms-25-12892-f001:**
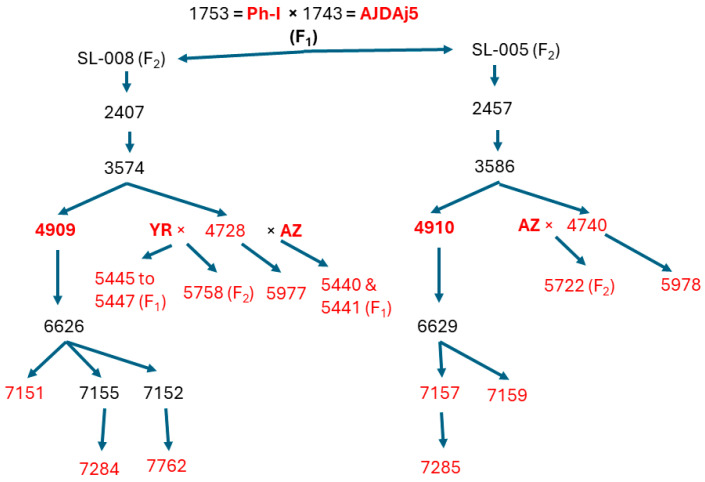
Pedigree of plant materials used in this study (in red color).

**Figure 2 ijms-25-12892-f002:**
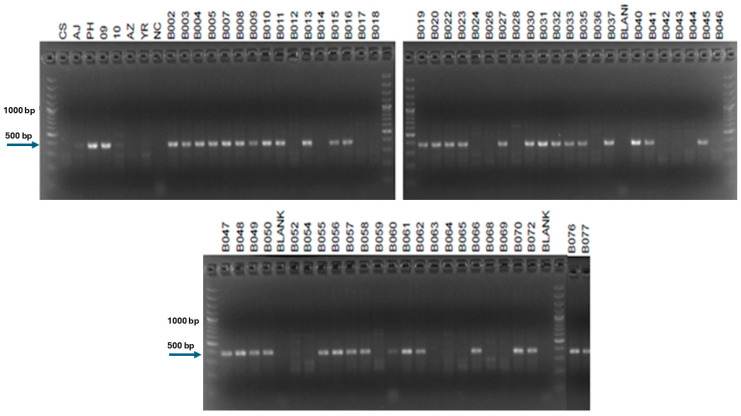
PCR amplification of RFLP-derived STS marker psr1205 (362 bp, arrowed) with the template DNA of Chinese Spring (CS), AJDAj5 (AJ), Ph1 inhibitor line (PH), 7151 = progeny of W4909 (09), 7157 = progeny of W4910 (10), Anza (AZ), Yecora Rojo (YR), negative control (NC), and F_2_ individuals (B numbers) of the cross YR × 4728-x (a sib of W4909).

**Figure 3 ijms-25-12892-f003:**
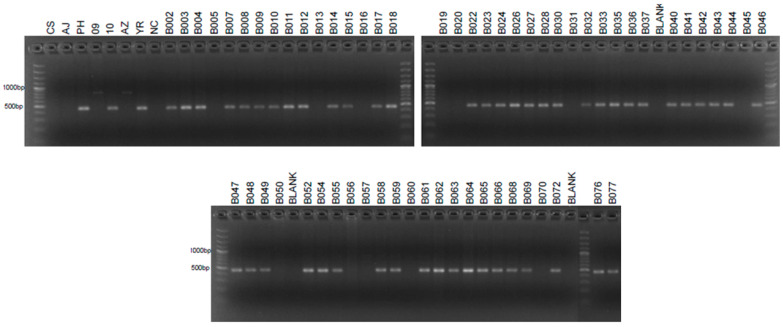
PCR amplification of SLAF-derived STS Marker264410 (462 bp) with template DNA of Chinese Spring (CS), AJDAj5 (AJ), Ph1 inhibitor line (PH), 7151 = progeny of W4909 (09), 7157 = progeny of W4910 (10), Anza (AZ), Yecora Rojo (YR), negative control (NC), and F_2_ individuals (B numbers) of the YR × 4728-x (a sib of W4909) cross.

**Figure 4 ijms-25-12892-f004:**
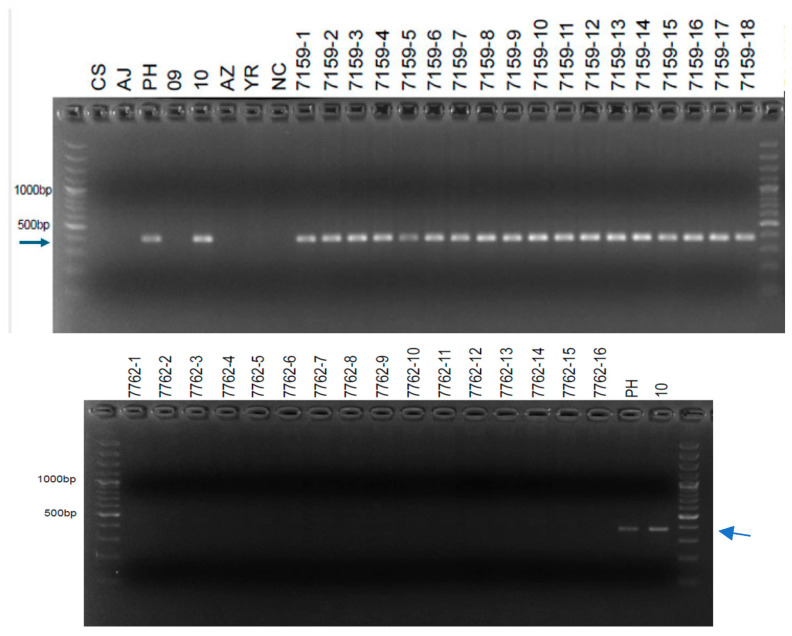
PCR amplification of SLAF-derived STS Marker6805321 (377 bp, arrows) with template DNA of Chinese Spring (CS), AJDAj5 (AJ), Ph1 inhibitor line (PH), 7151 = progeny of W4909 (09), 7157 = progeny of W4910 (10), Anza (AZ), Yecora Rojo (YR), negative control (NC), and individuals of lines 7159 (**top**) and 7762 (**bottom**), progeny lines from the lineage of W4910 and W4909, respectively.

**Figure 5 ijms-25-12892-f005:**
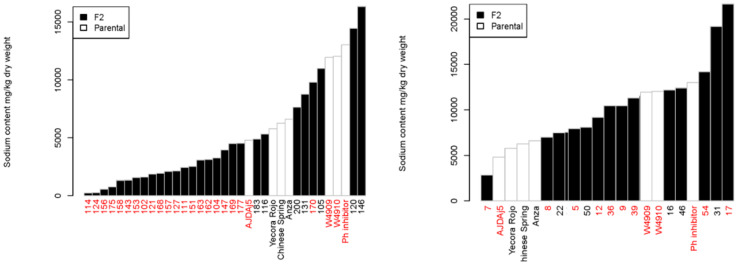
Leaf sodium concentration in F_2_ individuals of the crosses Anza × 4740 (**left**) and Yecora Rojo × 4728 (**right**). Plants with red identifications are salt-tolerant. Parental lines are represented by white bars.

**Table 1 ijms-25-12892-t001:** Analysis of SLAF sequences in Chinese Spring, AJDAj5, Ph-I, W4909, and W4910.

SNP Type	Number of SNPs	Number of SLAF Markers with ≥6 SNPs
Different between AJDAj5 and Ph-I	125,984	2030
Common between W4909 and W4910	73,326	218
Only in W4909	16,244	318
Only in W4910	19,464	456
Common between W4909 and W4910, and tracible to AJDAj5 and/or Ph-I	20,531	40
Total	255,549	3062

**Table 6 ijms-25-12892-t006:** Mean and 95% confidence interval (CI) of leaf sodium concentration of parental lines. Means with the same letter are not significantly different at *p* = 0.05 level.

Plant Materials	Control at EC = 0.6 dS/m		Salt-Treated at EC = 24 dS/m	
Name (ID Number ^1^)	Mean	95% CI	Mean	95% CI
Chinese Spring	1858 ^b^	1170 to 2546	6262 ^a^	4623 to 7901
(6687)				
AJDAj5	475 ^a^	222 to 728	4801 ^a^	3002 to 6600
(6598)				
Ph inhibitor	4287 ^c^	2717 to 5857	13,016 ^b^	9821 to 16,211
(6621)				
W4909	5818 ^c^	5275 to 6561	11,943 ^b^	10,647 to 13,239
(7151)				
W4910	5450 ^c^	3911 to 6989	12,030 ^b^	10,613 to 13,447
(7157)				
Anza	2083 ^b^	1513 to 2653	6610 ^a^	4746 to 8474
(5975)				
Yecora Rojo	1479 ^b^	1258 to 1700	5793 ^a^	4198 to 7388
(5976)				

^1^ ID numbers are seed packet numbers.

**Table 7 ijms-25-12892-t007:** Segregation of salt tolerance in F2 populations of Anza × 4740 and Yecora Rojo × 4728-x.

ID ^1^	Cross	No. Plants	SDST ^2^	No. Plants	Segregation Ratio	Chi-Square	*p*
5722	Anza × 4740	95	>52	67	3 : 1	1.0140	0.3139
			51.8 (50 to 52)	28			
5758	Yecora Rojo × 4728-x	66	>52	29	7 : 9	0.0011	0.9735
			49.0 (42 to 52)	37			

^1^ ID numbers are seed packet numbers. ^2^ SDST = survival days since treatment with NaCl salt; >52 = salt-tolerant; ≤52 = salt-sensitive.

## Data Availability

The original contributions presented in this study are included in the article/Appendix A; further inquiries can be directed to the corresponding author.

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
