# Peer review of "Dual Mechanisms of Salinity Tolerance in Wheat Germplasm Lines W4909 and W4910"

_ijms, 2024, doi:10.3390/ijms252312892_

Round 1
Reviewer 1 Report
Comments and Suggestions for Authors
Review:
Richard R.-C. Wang, Steven S. Xu, Thomas A. Monaco and Matthew D. Robbins
Dual mechanisms of salinity tolerance in wheat germplasm lines W4909 and W4910
The actuality of the choice of topic is given by the fact that in many countries of the world salt stress is the primary limitation of agricultural production. And wheat plays a significant role in food production, which makes the investigations of the working group important.
The research team used modern methods and techniques to explore the characteristics of the tested plant germplasm lines related to salt tolerance.
The investigations were thorough and extensive, the results are correct, and the conclusions drawn from them are reasonable.
I would like to make some additional comments to improve the quality of the publication:
1. The literary references used are relevant, although their number is few and most of them are not from the latest references. It would be good if more and timely publications were added to the manuscript.
2. It is not clear why the plants were planted in sand and then sprinkled with salt water, when it was mentioned that I quote: "The State of California in U.S. has a vast acreage of saline soil. Wheat cultivars Yecora Rojo (YR) and Anza ( AZ) are well adapted to California wheat production.
Why wasn't saline soil used for the tests?
3. The 2.2. there was a typo in the subsection. The correct expression is homozygosity.
4. In line 313, it is mentioned that the seeds were kept at 35°C in a refrigerator. Is it that correct? Either the seeds were kept in an air-conditioning chamber, a drying cabinet, or a thermostat, or the temperature given is incorrect.
5. In line 122, they mention that the distance between their markers and the gene was 40.33 centimorgans. This is considered a fairly large distance when using genetic maps. This possible reason is not mentioned when drawing conclusions, even though it may explain some of the results.
After considering the suggestions and implementing the amendments, I recommend the manuscript for publication.

Author Response
Reviewer#1’s comments
The actuality of the choice of topic is given by the fact that in many countries of the world salt stress is the primary limitation of agricultural production. And wheat plays a significant role in food production, which makes the investigations of the working group important.
The research team used modern methods and techniques to explore the characteristics of the tested plant germplasm lines related to salt tolerance.
The investigations were thorough and extensive, the results are correct, and the conclusions drawn from them are reasonable.
Response: Thank you for your positive comments upon the evaluation of our manuscript.
I would like to make some additional comments to improve the quality of the publication:
- The literary references used are relevant, although their number is few and most of them are not from the latest references. It would be good if more and timely publications were added to the manuscript.
Response: Thank you for this constructive comment. We added the following publications as [1, 2]
Tarolli, P.; Luo, J.; Park, E.; Barcaccia, G.; Masin, R. Soil salinization in agriculture: Mitigation and adaptation strategies combining nature-based solutions and bioengineering. iScience, 2024, 27, 108830. doi.org/10.1016/j.isci.2024.108830
Kamran, M.; Parveen, A.; Ahmar, S.; Malik, Z.; Hussain, S.; Chattha, M.S.; Saleem, M.H.; Adil, M.; Heidari, P.; Chen, J.T. An Overview of Hazardous Impacts of Soil Salinity in Crops, Tolerance Mechanisms, and Amelioration through Selenium Supplementation. Int J Mol Sci. 2019 21(1):148. doi: 10.3390/ijms21010148. PMID: 31878296; PMCID: PMC6981449.
and changed the [5] with
Koo, D.-H.; Friebe, B.; Gill, B.S. Homoeologous Recombination: A Novel and Efficient System for Broadening the Genetic Variability in heat. Agronomy 2020, 10, 1059. https://doi.org/10.3390/agronomy10081059
- It is not clear why the plants were planted in sand and then sprinkled with salt water, when it was mentioned that I quote: "The State of California in U.S. has a vast acreage of saline soil. Wheat cultivars Yecora Rojo (YR) and Anza ( AZ) are well adapted to California wheat production.
Why wasn't saline soil used for the tests?
Response: Silicate sand was used to ensure that plants for the control were grown in a medium having an EC as close to 0 dS/m as possible and plants for salt treatment were grown in that having an EC level set for the experiment. Saline soil from a field couldn’t meet this requirement.
- The 2.2. there was a typo in the subsection. The correct expression is homozygosity.
Response: Thank you for your careful checking on the spelling. It has been corrected.
- In line 313, it is mentioned that the seeds were kept at 35°C in a refrigerator. Is it that correct? Either the seeds were kept in an air-conditioning chamber, a drying cabinet, or a thermostat, or the temperature given is incorrect.
Response: Thank you for noticing this mistake. It was supposed to be 35°F, so we changed this to 2°C.
- In line 122, they mention that the distance between their markers and the gene was 40.33 centimorgans. This is considered a fairly large distance when using genetic maps. This possible reason is not mentioned when drawing conclusions, even though it may explain some of the results.
Response: We added this paragraph to the end of Discussion section (lines 310 to 314). “The genetic distance between molecular markers psr1205 and Marker264410 was fairly large, 40.33 cM. This fact could explain why the marker psr1205 was easily eliminated from the offsprings of W4909 (Table 5). The Su1-Ph1 gene linked to psr1205 can lead to chromosome instability, resulting in low seed-set. By selecting high seed yield in hybrid progenies of W4909, psr1205 and Su1-Ph1 gene could be eliminated.”
After considering the suggestions and implementing the amendments, I recommend the manuscript for publication.
Reviewer 2 Report
Comments and Suggestions for Authors
Major concerns
1. It is not recommended to merge gel pictures in Figure 4. There should be a marker line and positive control in the right part of Figure 4.
2. The color bars should be described in table 3, 4 and 5. Besides, where is the results of marker 4607154?
3. The amounts of plants for survival rate analysis seemed to be relatively small, which might affect the results. The authors should discuss it.
4. It is not enough to judge salinity tolerance by sodium ion concentration without biomass. How to evaluate the salinity tolerance if the material with low sodium concentration has a low biomass? The authors should try to provide precise standards for the evaluation.
Author Response
Reviewer#2 comments
Major concerns
- It is not recommended to merge gel pictures in Figure 4. There should be a marker line and positive control in the right part of Figure 4.
Response: Thanks for this suggestion. We have rerun the PCR and gel electrophoresis of the Marker6805321 with individuals of line 7762 along with positive controls, Ph-I and 7157 (progeny of W4910). The gel image is now the bottom part of Figure 4.
- The color bars should be described in table 3, 4 and 5. Besides, where is the results of marker 4607154?
Response: Thank you for this comment. We provided explanations for colored blocks or the + symbols in Tables 3, 4, and 5.
We already provided the reason for not presenting the result of Marker4607154 on lines 113 to 115 – “Results of PCR amplication for Marker4607154 in parental lines and hybrid progenies were same as those of Marker264410. Therefore, only the results of Marker264410 were presented.”
- The amounts of plants for survival rate analysis seemed to be relatively small, which might affect the results. The authors should discuss it.
Response: Originally, we planted 100 seeds of F2 of Yecora Rojo × 4728 in the 100 cups. When we obtained the gel image of psr1205, we noted that plants 78 to 100 were all negative for this marker. Then, we realized that plants 1 to 77 were grown from one packet of seed while plants 78 to 100 were from another packet, meaning that they derived from two F1 hybrids where the parent 4728-x was heterozygous for the marker psr1205. Therefore, results on plants 78 to 100 were removed from the data. This change didn’t significantly affect the chi-square test of the segregation ratio 7:9.
The 66 established F2 plants in the YR × 4728-x cross were more than enough to meet the minimum number of plants required to test a 2-genes F2 population. The equation for calculating minimum number of plants is 4n x 4, where n is the number of gene segregating. Thus, for a two-gene model the minimum number of F2 plants would be 42 x 4 = 4 x 4 x 4 = 64. Sixty-six plants were just more than enough to test the segregation ratio of 7:9.
- It is not enough to judge salinity tolerance by sodium ion concentration without biomass. How to evaluate the salinity tolerance if the material with low sodium concentration has a low biomass? The authors should try to provide precise standards for the evaluation.
Response: We agree with your comment. In this study, we observed that some single-tiller plants were salt tolerant while multi-tillered plants were salt sensitive. Therefore, biomass wasn’t a reliable criterion for salt tolerance. We used survival days since salt treatment (SDST) as the standard for evaluation of salt tolerance. It worked out quite well.
Round 2
Reviewer 2 Report
Comments and Suggestions for Authors
The authors have addressed most of the comments. The manuscript could be accepted at the current form.